# Stereotactic Body Radiotherapy for Patients with Lung Oligometastatic Disease: A Five-Year Systematic Review

**DOI:** 10.3390/cancers13143623

**Published:** 2021-07-20

**Authors:** Guillaume Virbel, Clara Le Fèvre, Georges Noël, Delphine Antoni

**Affiliations:** Department of Radiation Oncology, Institut de Cancérologie Strasbourg Europe (ICANS), 17 Rue Albert Calmette, BP 23025, 67033 Strasbourg, France; g.virbel@icans.eu (G.V.); c.lefevre@icans.eu (C.L.F.); d.antoni@icans.eu (D.A.)

**Keywords:** lung, metastases, oligometastatic, stereotactic body radiotherapy

## Abstract

**Simple Summary:**

Stereotactic body radiation therapy (SBRT) is currently used to treat lung metastasis, particularly in the setting of oligometastatic disease. Objective of this treatment is lesion’s ablation. However, its techniques, efficacy, and safety remain to be evaluated. Several teams have published their experiences with different radiotherapy schedules. The aim of this review was to analyze these topics in regard to the last five years of publications. For this purpose, we carried out a systematic review of the literature using the PRISMA method. This review can help oncologists involved in lung oligometastasis to clarify their knowledge through the wealth of published data and may lead to improved management.

**Abstract:**

For several years, oligometastatic disease has represented an intermediate state between localized disease accessible to local treatment and multimetastatic disease requiring systemic therapy. The lung represents one of the most common metastatic locations. Stereotactic body radiation therapy (SBRT) appears to be the treatment of choice for these patients. There are few data defining the place of radiotherapy and reporting outcome after SBRT in lung metastases. This 5-year review aimed to determine areas of SBRT usefulness and methods for the management of pulmonary metastasis in oligometastatic patients. A search for articles on PubMed allowed selection of the most relevant studies. Eighteen articles were selected according to pre-established criteria for this purpose. The analysis concludes that SBRT is an effective and safe treatment in selected patients when the disease remains localized from one to three organs.

## 1. Introduction

For several years, Hellman and Weichselbaum described oligometastatic disease evolution as involving a low number of metastases in only a few organs [1]. Depending on the authors, the definitions have varied from one to five metastases in one to three sites [2,3,4]. Some studies have suggested that life expectancy could be longer for these patients compared with those with more spread out disease [5,6]. Moreover, some studies have shown that local treatments can improve overall survival [2,7]. SABR-COMET, a recent phase II study, investigated the effect of stereotactic body radiation therapy (SBRT) on oligometastatic patients’ survival. SBRT was associated with an improvement in overall survival compared with palliative standard of care treatments alone [2].

The lung is one of the most frequent sites of metastases for solid tumors [8]. Lung metastasectomy is the historical treatment, but it requires good general conditions and cardiovascular and respiratory functions [9,10,11]. Based on the favorable results of SBRT in non-small cell lung cancer (NSCLC) [12,13,14], it was proposed as a curative treatment for patients with lung metastases that were unsuitable for surgery. However, patient selection criteria remained blunt, and several variable irradiation schedules were used. No robust data allowed recommendations, although reported local control and survival rates appeared encouraging. We led a review of the last five years of published articles to assist in designing useful guidelines for SBRT modalities.

## 2. Materials and Methods

Article research followed the “PRISMA method” [15]. Articles corresponding to the terms “lung metastases” and “SBRT” were searched in the PubMed database (https://www.ncbi.nlm.nih.gov/pubmed; accessed on 31 December 2020). Studies published since 2015 reporting on case series published in English or French were included. We closed the search on 31 December 2020. The selected studies only reported patients treated for lung metastasis with SBRT, regardless of the primitive cancer. Only case series including at least 20 patients were retained for the analysis. In each article, we collected the number of patients, their gender, age, number of irradiated pulmonary metastases, period of inclusion, the “oligometastatic” definition used by the authors, size of the lesions, primary tumor, metachronous or synchronous nature of the lesions, exclusive lung oligometastatic state, complications rate, and follow-up time, as well as local control (LC) and overall survival (OS) or progression-free survival (PFS) rates. To be selected, an article had to absolutely mention the prescribed dose, prescription isodose, definition of target volumes, and at least 70% of the previously listed items (Appendix A).

Overall, 279 articles were retrieved. Among them, 261 articles were excluded because they did not meet the inclusion criteria. Ultimately, 18 retrospective studies were included. Among them, six focused exclusively on colorectal lung metastases and three on sarcoma lung metastases (Figure 1).

## 3. Results

### 3.1. Patient Population

Eighteen studies reported data on 1191 patients. The M/F sex ratio was 1.5 (677 males and 470 females). The median age varied from 56 to 74 years old. The ECOG index ranged from 0 to 3, with only one study reporting ECOG for three patients [16]. Oligometastatic disease was defined as one to five metastases in eight articles [17,18,19,20,21,22,23,24], one to four metastases in one article [25], one to three metastases in three articles [26,27,28], and six articles did not specify any definition [16,29,30,31,32,33].

The number of irradiated metastases was 1705. The median number of metastases per patient ranged from one to two. The tumor diameters varied from 2 mm [17] to 124 mm [28]. Eight articles described local treatment prior to SBRT. Surgery was used at least once before SBRT in 164 patients, but no thermoablation before SBRT was identified. Only two authors specified surgery for metastases, which represented 44 patients [18,21,22,24,25,27,30,33]. The 2-year LC, PFS, and OS ranged from 31% to 93%, from 10% to 38%, and from 40% to 85%, respectively. The characteristics and outcomes of patients are reported in Table 1.

### 3.2. Primary Cancer and Time to SBRT

The primitive tumor was localized in the lung or colorectal tract for 572 and 194 patients, respectively. All irradiated lesions were metastases. These two locations represented 70% of the primaries. Helou et al. described 26 metastases from lung cancer and 101 from colorectal cancer [19]. Metastatic disease reached the lungs in 803 patients but spread to various other locations in 182 patients; data were unavailable for 206 patients. SBRT was delivered for synchronous metastases in 170 patients and for metachronous metastases in 251 patients [17,18,20,21,22,25,26,32]. Two authors reported 161 synchronous metastases and 298 metachronous metastases [23,24]. In 14 articles, the authors reported chemotherapy before SBRT, which was delivered to 386 patients [16,17,18,20,21,22,24,25,26,27,29,30,31,33]. Sharma et al. reported patients treated for 159 metastases who received systemic therapy [23].

In only two articles, the authors divided oligometastatic and oligoprogression patients. Li et al. reported 13 patients (24.5%) with oligoprogression disease [22]. In Helou et al., 38 lesions (21%) were oligoprogressive metastases [19].

### 3.3. SBRT Technique

#### 3.3.1. Set-Up

Six studies, for a total of 386 patients (32%), reported fiducial implantation for all [23,28,29,31] or some [24,30] of the patients. Fiducial markers were also used in 13 metastases of the 33 (39%) reported by Bauman et al. The authors did not explain the reason for using this technique for these patients [30]. A 4D scanner was used in 478 patients (44%) [16,17,21,22,25,26,27,33]. Two studies reported 26 (67%) and 45 (76%) lesions delineated from 4D scanner slices [20,30]. An abdominal compression technique during the CT acquisition was used in 164 (15%) patients [18,19]. Filippi et al. used this technique for 14 (24%) metastases [20], and Pasqualetti et al. used a three-phase scanner for the 33 patients [32]. All patients were treated in the supine position.

#### 3.3.2. Radiation Therapy Prescription

One study reported a 30 Gy monofractionated irradiation for all patients, corresponding to a biological equivalent dose (BED) of 120 Gy (considering α/β = 10 Gy) [17]. Other authors reported doses ranging from 18 to 75 Gy in 1 to 10 fractions corresponding to a 10GyBED ranged from 43.2 to 120 Gy. The prescription isodoses varied from 70% to 95%. Eight studies described specific schedules for lesions located in the no-fly zone [18,19,21,23,25,27,28,30]. Doses ranging from 26 to 60 Gy in 1 to 5 fractions were used for peripheral metastases, while central lesions received 18 to 60 Gy in 1 to 8 fractions. Gross tumor volume (GTV) was delineated on planning CT. Two teams matched the diagnostic PET/CT images with the planning CT [17,25]. An internal target volume (ITV) was defined in nine studies [17,18,19,21,22,25,26,27,33]. For five studies, the clinical target volume (CTV) corresponded to the GTV [20,21,27,28,32]. One study reported a 3 mm margin CTV [24]. In the other studies, CTV was not defined [16,17,18,19,22,23,25,26,29,30,31,33]. Planning target volume was defined with a 3 to 5 mm margin from the GTV or CTV. Only Qiu et al. reported a higher PTV margin, from 7 to 11 mm [16]. Filippi et al. used a changing margin according to the use or not of the VMAT technique, with 3 or 10 mm, respectively [20].

#### 3.3.3. Technique of Irradiation

A total of 73 lesions (4%) were treated with three-dimensional-conformal radiotherapy (3D-CRT), 229 with volumetric-modulated therapy (13%), 271 with intensity-modulated radiotherapy (IMRT) (16%), and 675 with a cyberknife (40%). Five studies did not report the irradiation technique, representing 457 lesions (27%) [16,22,25,32,33].

### 3.4. Follow-Up and Evaluation after SBRT

#### 3.4.1. Evaluation

All series used a CT scan to follow up on the lesion evolution. Metabolic imaging was used in 14 articles [17,18,20,21,22,23,24,25,26,27,29,30,31,32]. Only four authors reported histological confirmation in case of local failure [22,23,27,30].

#### 3.4.2. Outcome

The average follow-up ranged from 13 to 43 months after radiotherapy. The 2-year LC, PFS, and OS extended from 31% to 93%, 10% to 38%, and 40% to 85%, respectively. Nine studies reported a 2-year LC greater than 80% [17,18,19,21,22,23,27,29,30]. Among 1191 patients, 79 patients (6.6%) developed a local failure [17,18,20,21,25,26,27,28,29,30,33], representing 118 metastases among 1705 (6.9%) [17,19,21,22,25,30,32,33]. Ten studies reported that 276 patients (49%) developed distant metastases [17,18,20,21,22,25,26,27,29,33]. Four authors described 67 patients who received a second SBRT for new lesions [20,22,24,33].

#### 3.4.3. Complications

Grade 3 acute pneumonitis [17,19,24,28,29], late lung fibrosis [17,24,31], rib fracture [17], and esophagitis [33] were developed in 12, 14, 2, and 1 patients, respectively. One case of grade 4 atelectasis [31], one case of grade 4 late lung fibrosis [24], and two cases of grade 5 toxicity were reported [17,19]. Ten studies did not find any grade 3 or higher toxicity [16,18,20,21,22,25,26,27,30,32].

Only one study showed a relationship between the modalities of SBRT and toxicities. A dosimetric analysis revealed that the prescription maximum point (51.9 vs. 47.4 Gy) and higher bronchus maximum point dose (49.0 vs. 46.1 Gy) were statistically significant predictors of late grade 2 or higher atelectasis [31].

### 3.5. Prognostic Factors

Prognostic factors are reported in Table 2.

#### 3.5.1. Demographic Factors

Age, gender, performance score before SBRT, and tumor location did not have any impact on the rate of LC [16,17,21,22,24,26,27,29,30].

#### 3.5.2. Tumor Size

LC was significantly better for the smallest metastases in three series [17,18,19]. Osti et al. and García-Cabezas et al. found that LC rates were significantly better for metastases smaller than 18 and 25 mm, respectively; the 5-year LC was 89.5% vs. 55.3%, (*p* = 0.001) in Osti et al. [17] and 100% vs. 78.9% (*p* = 0.023) in García-Cabezas et al. [18]. Helou et al. found that the largest lesions were independently associated with worse LC (HR 1.45; 95% CI: 1.01–2.10; *p* = 0.046). These authors did not specify a size threshold [19]. In contrast, Wang et al. did not find any LC differences between the volume of lesions more or less than 10 cc [29].

#### 3.5.3. Number of Thoracic Metastasis Lesions

In univariate analysis, Wang et al. reported that the number of thoracic metastases was correlated with PFS. The 2-year PFS rate was 33.4% for patients with one lung metastasis compared with 0% for the other patients (*p* = 0.012) [29].

#### 3.5.4. Biological Effective Dose (BED)

Three series reported that a 10GyBED of 100 Gy was a LC significant predictive factor, as in Wang et al. (*p* = 0.02) [29]. Sharma et al. showed that this BED allowed a 4-year LC of 87% vs. 64%, with a lower BED (HR 3.59; 95% CI: 2.00–6.44; *p* < 0.01) [23]. Comparably, Helou et al. reported that a lower BED was independently associated with an increase in local failure (HR 0.96; 95% CI: 0.93–0.99; *p* = 0.02) [19]. A 10GyBED of 120 Gy was the only LC prognostic factor in Berkovic et al. [24]. Figure 2 shows local control rates as a function of 10GyBED (Figure 2).

#### 3.5.5. Fractionation of SBRT

In their univariate analysis, Sharma et al. found that a single-fraction irradiation was associated with lower LC than a multifractionated irradiation (HR 2.83; 95% CI: 1.43–5.59, *p* = 0.003). This result was not found in multivariate analysis [23].

#### 3.5.6. Time to SBRT

In studies reporting this information, patients with synchronous metastases tended to have poorer survival. Lee et al. showed a lower OS rate in this subgroup compared with patients treated for metachronous metastases (*p* = 0.026) [28]. Sharma et al. demonstrated comparable results, with a hazard ratio of 2.21 (95% CI: 1.22–4.00; *p* = 0.009) in favor of patients with metachronous metastases [23]. Li et al. found more regional metastases at one year in the oligoprogression subgroup than in the oligometastatic group (79.5% vs. 25.1%; *p* = 0.001) [22].

#### 3.5.7. Combination with Chemotherapy

Sharma et al. reported that patients who received pre-SBRT chemotherapy had lower local control (HR 2.61; 95% CI: 1.46–4.64; *p* = 0.001). The authors hypothesized that this could be due to accelerated repopulation of clonogenic tumor cells, where the surviving tumor cells may develop more resistance and may have acquired better DNA repair capacity after the initial exposure to chemotherapy [23].

#### 3.5.8. Primary Colorectal Cancer

For Osti et al., the colorectal primary of lung metastasis (CRLM) was a significant predictive factor of poor LC and PFS. The 3-year LC rates were 64.8% for patients with CRLM compared with 86.3% for other primary tumors (*p* < 0.01), and the 3-year PFS rates were 12.1% and 41.9%, respectively (*p* = 0.03) [17]. Helou et al. reported a hazard ratio of 13.59 (95% CI 4.19–44.12; *p* < 0.001), with the poorest LC rates for CRLM [19].

### 3.6. Prognostic Factors of Specific Histologies

#### 3.6.1. Colorectal

Seven articles including 344 patients with 169 metastases studied the outcome of the colorectal cancer subgroup [16,19,20,22,25,26,32]. Delivered doses were largely studied. Agolli et al. showed that BED > 100 Gy led to a significantly better 1-year LC (HR 0.20; 95% CI: 0.04–0.98; *p* = 0.047) [25]. For Helou et al., a total dose of 60 Gy was associated with a significantly lower HR of local failure (HR 0.22; 95% CI 0.07–0.75; *p* = 0.015) [19]. Jung et al. found a non-significant trend toward a lower 3-year LC rate when lesions received a total dose less than 60 Gy [26]. CLRM local control according to the BED was below those of other primitives (Figure 2).

Jung et al. showed that a pre-SBRT carcinoembryonic antigen was significantly predictive of the LC rate. Three-year LC rates were 0% and 78.1% for patients with CEA < 6 µg/L and ≥ to 6 µg/L, respectively (*p* < 0.01) [26].

A larger tumor size was also predictive of poorer LC [16,19,22,26] and OS [26]. Helou et al. reported an HR of 1.85 (95% CI: 1.19–2.88; *p* = 0.006) in favor of a smaller size [19]. A GTV less than 1.5 mL or 1.6 mL was associated with a 1-year LC of 98% or a 3-year LC of 88.5% compared with 84% (*p* = 0.011) and 50.1% (*p* = 0.01), respectively, if GTV was larger [22,26]. Qiu et al. showed that patients who received SBRT for a lesion with a GTV greater than 14.1 mL had worse LC (*p* = 0.02) [16].

Jung et al. found that the timing of SBRT (for a second or third lung metastasis) was a statistically significant prognostic factor for 3-year LC, with 63% for the first lung metastasis and 91% for the others (*p* = 0.04). The first session may have selected “good responder” patients for whom a second or even a third session was more easily indicated [26]. However, the authors did not discuss the possibility that the radiosensitivity of the first treated metastasis could be a marker for irradiating subsequent metastases that remained radiosensitive. For Li et al., patients treated for synchronous oligometastatic diseases or multiple lung metastases had a worse PFS than patients with a metachronous or single lesion (6% vs. 46%; *p* < 0.001 and 11% vs. 44%; *p* = 0.04, respectively) [25]. Li et al. found that oligoprogression was a worse predictor of regional control compared with oligometastatic disease (HR 2.78; 95CI: 1.04–7.48; *p* = 0.042) [22].

#### 3.6.2. Sarcomas

Three articles studied SBRT specifically for lung metastases from sarcomas [21,30,33]. This represented 102 patients and 207 metastases. For Navarria et al., the disease-free interval (DFI) from diagnosis significantly affected survival; the 3- and 5-year OS rates of patients with DFI ≤ 24 months and those with DFI > 24 months were 88% vs. 67% and 88% vs. 33%, respectively (*p* = 0.02) [21].

## 4. Discussion

For non-operable patients, to control lung metastases, SBRT seems to be the best option. Indeed, in recent years, technical progress has allowed delivery of an ablative dose to various locations [34]. Although no phase III trials are specifically available to definitively prove the efficiency of SBRT, several retrospective studies have demonstrated relevant results in lung oligometastasis [35,36].

Main objective of SBRT is the lesion’s ablation. LC and survival data was good in our study, as in other reviews [37]. Reported LC rates are noteworthy, reaching up to 96% according to Navarria et al. [21]. However, the technical modalities are not yet standardized. Radiation oncologists have used different schedules according to the previously published results, mainly mimicking the treatment of stage I primitive lung cancer or following their own experiences. This heterogeneity has induced difficulties in comparing the studies and, consequently, proposing standard treatment protocols.

To our knowledge, this PRISMA literature review is the largest study regarding lung SBRT for oligometastatic lesion. More than one-third of patients reported in our analysis had a fiducial implantation. Neither of the two studies including patients treated with or without fiducial implantation reported an advantage to this technique [24,30]. Local control reported in studies with patients with fiducial implantation was not greater than that in studies with patients without fiducial implantation (57.4% to 90.6% vs. 30.9% to 96%) [23,28,29,31]. In addition, the pneumothorax risk was measured from 9% to 23%, depending on the study and this technique delays treatment [24,38]. It is therefore not appropriate to recommend systemic fiducial implantation as a tracking technique.

More than half of patients in our analysis had received a 4D scan. Although there is no demonstrated advantage, the use of a 4D scanner seems to be the best non-invasive technique to integrate respiratory movements to define the ITV [39].

The optimal physical dose to reach a high probability of LC is unknown, but this review suggests that a 10 GyBED of 100 Gy could be proposed to increase the LC [19,23,25,29]. This result is parallel with data obtained for NSCLC in which a 100GyBED affected LC [13,40]. CRC metastases appear to require a larger BED. The only prescription dose prognostic factor of LC found in the series concerned CRLM. Indeed, 60 Gy in 4 or 5 fractions could be a relevant threshold (corresponding to a BED of 132 or 150 Gy) [19,26]. Mazzola et al. found better result of SBRT after bevacizumab in lung oligometastases from colon cancer [41].

A multi-institutional phase I/II trial assessed SBRT for lung metastases. The delivered irradiation dose varied from 48 to 60 Gy in 3 fractions. This corresponded to a 10GyBED from 106 to 120 Gy. The actual 1- and 2-year LC rates were 100% and 96%, respectively [42].

Complication rate did not differ significantly by fractionation in studies reporting single and polyfractionated treatment [20,21,23,25,27,29]. However, care must be taken when irradiating lesions located in the no-fly zone or near the chest wall. Indeed, an adapted fractionation is necessary during the treatment of the most central lesions in order to obtain LC and toxicity similar to that of peripheral lesions [43,44]. Mazzola et al. proposed a simultaneous integrated protection (SIP) strategy for SBRT of central lung lesion with good results [45].

There was no prognostic factor to define the best margins in this review.

Patients with oligometastatic disease are eligible for curative treatment. If a cure cannot be achieved, the treatment of each lesion can improve overall survival and PFS [46,47].

LC could be a survival prognostic factor. Indeed, a retrospective, multicenter European study showed that lung metastasis removal led to encouraging long-term survival, specifically after complete resection. The 5-year survival rates after complete and partial metastasectomy were 36% and 13%, respectively [10]. The good local results reported in this review and the absence of difference in terms of CL comparing SBRT and surgery argue to propose SBRT as an ablative treatment of lung metastases that can improve patient survival [28]. In the SABR-COMET phase II study, OS was significantly better in the experimental group. The median OS was 28 months in the control arm vs. 50 months in the SBRT arm (*p* = 0.006). In this study, the lung was the most common metastatic site and was involved in 53% of patients in the control arm and 43% in the SBRT arm. The treatment schedule varied according to lesion location in lung parenchyma. Doses varied from 54 to 60 Gy in 3 to 8 fractions every second day [2]. COMET 3 and COMET 10 phase III trials comparing SBRT and palliative care may help to confirm SBRT place in treatment oligometastatic patient’s treatment.

Depending on the studies, the 2-year OS rates varied by 2-fold (40 to 85%) [21,31]. This difference reflects heterogeneity in the patient population included in the studies. Indeed, all studies were retrospective, and the first inclusion criterion was to have received SBRT for oligometastatic disease. A better selection of patients would be necessary to obtain better results. However, currently, there are no well-established criteria for this selection.

The median number of metastases per patient reported in this review ranged from one to two, consequently, for oligometastatic patients, it seems reasonable to propose SBRT up to two lung lesions. However, no data are available to advise repetition of SBRT with this number of metastases.

Clinical factors may split patient outcomes and could be used to adapt radiation treatment schedules. The responses of synchronous, metachronous, and progressive tumors were different. Patients with synchronous metastases have poorer prognoses [22,23,28]. Alongi et al. concluded the same results in a previous literature review [37]. Reported variables in the used series were so heterogeneous that it made our analysis difficult. The primary tumors were variable, and irradiation occurred at different times during the history of disease. In this context, EORT defines genuine oligometastatic disease with patients with no history of polymetastatic disease having low metastatic capacity. SBRT could delay the introduction of poorly tolerated systemic therapy. This would allow maintenance of a better quality of life. In contrast, some patients have an induced oligometastatic disease with a history of polymetastatic disease. These patients have a disease with metastatic potential. In this case, SBRT consolidates systemic therapy and aims to restore a status of stable disease or a status of complete response. It may also aim to restore a status of overall sensitivity to systemic therapy through eradication of oligometastases with resistance to the current line of systemic therapy [48]. A review of the literature has specifically studied the latter in the context of patients with oligoprogressive oncogene-addicted NSCLC treated with SBRT. The authors reported good results, with PFS of more than 10 months after local treatment and continued systemic therapy in patients with EGFR-mutated NSCLC considered resistant to TKIs [49].

The circulating micro-DNA analysis seems promising and could influence the oligometastatic or polymetastatic tumor profile phenotype through gene expression regulation [50,51]. This parameter could participate in the development of a predictive score for patients’ metastatic history, with histology, size of the lesion, lesion numbers, metastatic site numbers, time between initial diagnosis, and metastases appearance and could better characterize the type of oligometastatic disease the patient has in order to optimize treatment.

It is still necessary to define scores and criteria to select patients for whom SBRT is impactful given their disease history. A data base for oligometastatic patients would be useful in the absence of phase III trials. Radiomics and genomics are interesting approaches to improve the characterization of the patient’s prognosis [52].

It is reasonable to treat one or two lesions, while obtaining satisfactory toxicity and efficacy. This number could be increased if dosimetric conditions are respected to avoid any adverse effect; Lindsay et al. irradiated up to seven lesions per patient with a seemingly acceptable tolerance [33].

In a strategy to improve survival, toxicity must be evaluated. In this analysis of the literature, the most frequently reported complications were pneumonitis or lung fibrosis. Their incidences remained low, but several reasons can explain these results. First, given the reluctance to perform multiple exams in patients with metastatic disease, radiation complications can arise late and can thus be rarely diagnosed as a consequence of the relatively short life expectancy of the patients. However, the scarcity of these complications with SBRT had already been reported in NSCLC [53], and they are more rare than after conventional fractioned radiotherapy [54,55]. In NSCLC, the RP incidence (grade 2 or more) after SBRT varied from 2% to 28%, and this diagnosis was clinical and radiological [53]. None of the selected studies found a prognostic RP factor. A Danish study showed that comorbidity, older age, and average or lower location of the lesion influenced RP risk, but they also described tobacco as a protective factor. These data concern treatment with normofractioned radiotherapy [56]. Johansson et al. presented the same results about smokers, even with traditional splitting. They hypothesized that it could be explained by immunological reactions induced by cigarette smoke or by hypoxia, which smoking generates. However, the role of tobacco remains controversial [57]. Menoux et al. proposed recommendations concerning dosimetric parameters to be applied to reduce RP risk. They also point out that routine use of these parameters is difficult to apply, knowing that the definition of “healthy lung” varies depending on the study [53].

## 5. Conclusions

SBRT is an efficient and well-tolerated treatment for lung metastases in oligometastatic patients. However, the optimal treatment schedule is not definite. Protocols used for NSCLC, with a BED > 100 Gy, appear to be appropriate to obtain a LC comparable with that of surgery. It seems necessary to define the type of oligometastatic disease of the patients with the help of clinical, radiological, and biological markers in order to opt for the best strategy. SBRT is an ablative treatment to be integrated in this strategy, especially when surgery alone is unreasonable.

## Figures and Tables

**Figure 1 cancers-13-03623-f001:**
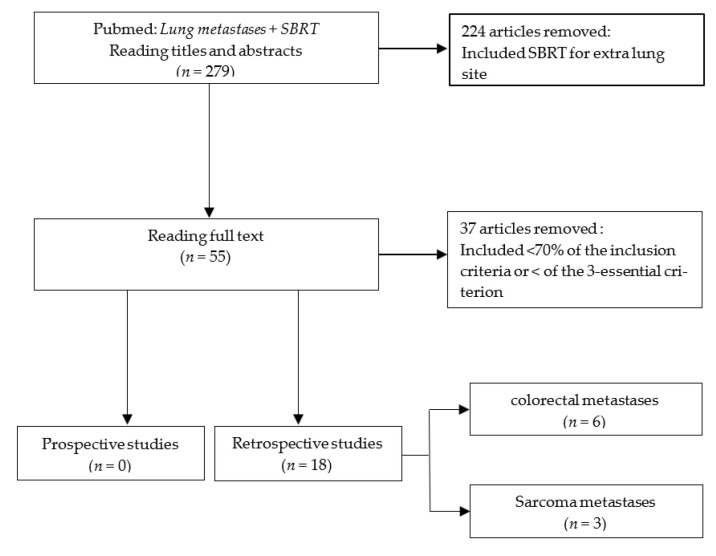
Flowchart of published selected articles.

**Figure 2 cancers-13-03623-f002:**
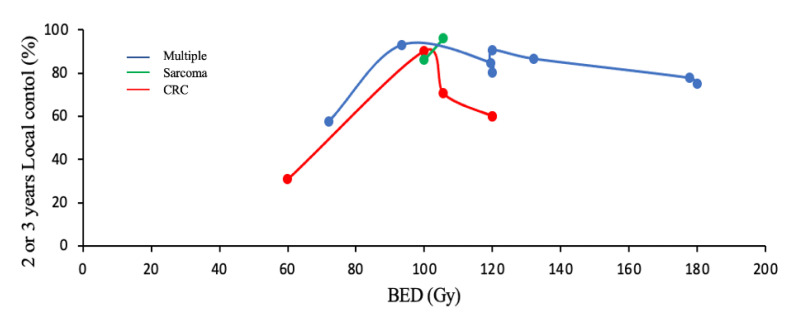
Local control depending on biologically effective dose according to the primitive cancer (considering α/β = 10 Gy).

**Table 1 cancers-13-03623-t001:** Characteristics and outcome of lung SBRT in the 18 studies.

Author	Number of Irradiated Metastases Per Patient	Treated Patient	Irradiated Metastases	Size of Lesions	Primitive Cancer	Dose	Local Control	PFS	Evaluation Imaging
Jung et al. 2015 [26]	1–3	50	79	1.5 cc (med)	CRC	40 to 60 Gy (med 48 Gy); 3 or 4 frct	70.6% (3 years)	24% (3 years)	PET/CT or CT
Garcia-Cabezas et al. 2015 [18]	-	44	53	2.0 cm (med)	multiple	50 to 60 Gy; 5 to 8 frct	86.7% (2 years)	ns	PET/CT
Filippi et al. 2015 [20]	1–4	40	59	1.5 cm	CRC	26 to 60 Gy; 1 to 8 fract	ns	27% (2 years)	PET/CT or CT
Navarria et al. 2015 [21]	1–4	28	51	6.5 cc (med)	sarcoma	30 to 60 Gy; 1 to 8 frct	96% (2 years)	ns	PET/CT or CT
Wang et al. 2015 [29]	1–4	95	134	14.6 cc (med)	multiple	30 to 60 Gy; 1 to 5 frct	90.6% (2 years)	29% (2 years)	PET/CT
Siva et al. 2015 [27]	1–3	65	85	ns	multiple	18 to 50 Gy; 1 to 5 frct	93% (2 years)	38% (2 years)	PET/CT
Lischalk et al. 2016 [31]	ns	20	20	85.8 cc (med)	multiple	35 to 40 Gy; 5 fract	57.4% (2 years)	ns	PET/CT or CT
Baumann et al. 2016 [30]	ns	30	39	2.4 cm (med)	sarcoma	50 Gy (med); 4 or 5 frct	86% (2 years)	ns	PET/CT or CT
Pasqualetti et al. 2017 [32]	1–3	33	56	2.3 cc (med)	CRC	24 to 42 Gy; 3 frct	62% (1 year)	10% (2 years)	PET/CT or CT
Agolli et al. 2017 [25]	1–4	44	69	1.4 cm (med)	CRC	23 to 45 Gy; 1 to 3 frct	60.2% (2 years)	16% (3 years)	PET/CT
Lindsay et al. 2018 [33]	1–7	44	117	2.1 cm (med)	sarcoma	50 Gy; 10 frct	ns	ns	CT
Qiu et al. 2016 [16]	ns	65	ns	ns	CRC	50 Gy; 5 or 10 frct	30.9% (2 years)	23.5% (1 year)	ns
Osti et al. 2018 [17]	1–5	129	166	1.3 cm (ave)	multiple	30 Gy; 1 frct	80.1% (3 years)	34% (3 years)	PET/CT or CT
Lee et al. 2018 [28]	1–3	21	29	2.5 cm (med)	multiple	60 Gy; 3 frct or 48 Gy; 4 frct	75.2% (2 years)	12 (2 years)	ns
Sharma et al. 2018 [23]	1–5	206	327	ns	multiple	30 to 60 Gy; 1 to 8 frct	85% (2 years)	36% (2 years)	PET/CT or CT
Li et al. 2019 [22]	1–4	53	105	1.1 cm/1.6 cc (med)	CRC	48 to 75 Gy; 4 to 10 frct	90.4% (1 year)	ns	PET/CT
Helou et al. 2017 [19]	ns	120	184	1.5 cm (med)	multiple	48 to 60 Gy; 4 to 5 frct	84.8% (2 years)	ns	CT
Berkovic et al. 2020 [24]	1–4	104	132	7.9 cc (ave)	multiple	20 to 60 Gy; 3 or 5	77.8% (3 years)	ns	PET/CT or CT

ave: average; CRC: colorectal cancer; frct: fraction; med: median; ns: not served; PFS: progression-free survival.

**Table 2 cancers-13-03623-t002:** Clinical prognostic factor according to the 18 studies.

Author	Tumor Size	Number of Thoracic Metastasis Lesions	Biological Effective Dose	Time to SBRT	Combination with Chemotherapy	Primary Colorectal Cancer
Jung et al. 2015 [26]	2.5 cm (LC)	-	-	-	-	-
Garcia-Cabezas et al. 2015 [18]	-	-	-	-	-	-
Filippi et al. 2015 [20]	-	-	-	-	-	-
Navarria et al. 2015 [21]	-	-	-	-	-	-
Wang et al. 2015 [29]	-	1 (PFS)	100 Gy (LC)	-	-	-
Siva et al. 2015 [27]	-	-	-	-	-	-
Lischalk et al. 2016 [31]	-	-	-	-	-	-
Baumann et al. 2016 [30]	-	-	-	-	-	-
Pasqualetti et al. 2017 [32]	-	-	-	-	-	-
Agolli et al. 2017 [25]	-	-	-	-	-	-
Lindsay et al. 2018 [33]	-	-	-	-	-	-
Qiu et al. 2018 [16]	-	-	-	-	-	-
Osti et al. 2018 [17]	1.8 cm (LC)	-	-	-	-	CLRM < no CLRM (LC and PFS)
Lee et al. 2018 [28]	-	-	-	Synchronous < metachronous (OS)	-	-
Sharma et al. 2018 [23]	-	-	100 Gy (LC)	Synchronous < metachronous (OS)	pre-SBRT CT < no pre-SBRT CT (LC)	-
Li et al. 2019 [22]	-	-	-	Oligoprogression < oligometatic (regional metastases)	-	-
Helou et al. 2017 [19]	largest lesions (LC)	-	100 Gy (LC)	-	-	CLRM < no CLRM (LC)
Berkovic et al. 2020 [24]	-	-	120 Gy (LC)	-	-	-

CLRM: colorectal primary of lung metastasis; CT: chemotherapy; LC: local control; OS: overall survival; PFS: progression-free survival.

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
