# Peer review of "Stereotactic Body Radiotherapy for Patients with Lung Oligometastatic Disease: A Five-Year Systematic Review"

_cancers, 2021, doi:10.3390/cancers13143623_

Round 1

Reviewer 1 Report

The authors have selected a series of papers published over a 5 year period using SBRT in the treatment of oligometastatic disease with lung metastases. They have selected 18 papers, all retrospectative series, including at least 20 patients regardless of the primary disease. They present in two table the information including in the papers and some results. If a reader is interested to save time then it show quickly what information is available in those papers but  as mentionned by the authors, the series are very heterogeneous and all papers are not included all the information. The management of an oligometastatic disease is a complicated issue and SBRT is only one comment. Clearly, we know already the efficacy of SBRT and the low complication rate.

In my view the two most interesting information is provided by the figure 2 and the comment on fiducial marker.

So, there are too many questions to consider the paper except if you wish to have a simple list of papers and in my personnel view, this is to weak to consider it fora publication

Reviewer 2 Report

The present review is interesting, however I suggest some editing of English language and style to improve clarity.

Here are my remarks:

Page 1, Line 30, section 1 Introduction: The Authors write "The lung is the secondary site of metastases for solid tumors". Probably the Authors mean that it is the most frequent site, or one of the most frequent.

Page 5, Line 109, section 3.3.2 Radiation therapy prescription: "Eight studies described specific schedules for lesions located in the flying zone". I think that the Authors aim to refer to the so-called "no-fly zone".
Page 6, Line 127, section 3.4.1 Evaluation: please correct "Metabolic imagery" in "Metabolic imaging"

Page 7, Line 176, section 3.5.5 Fractionation of SBRT. The Authors state that "In their univariate analysis, Sharma et al. found that a single-fraction irradiation was associated with lower LC than a multifractionated irradiation"; this can be due to the use of a too low dose in the single fraction cohort, other studies,for example RTOG 0915,  have shown comparable efficacy when choosing the appropriate dose (Alongi F, Nicosia L, Figlia V, De Sanctis V, Mazzola R, Giaj-Levra N, Reverberi C, Valeriani M, Osti MF. A multi-institutional analysis of fractionated versus single-fraction stereotactic body radiotherapy (SBRT) in the treatment of primary lung tumors: a comparison between two antipodal fractionations. Clin Transl Oncol. 2021 Apr 10. doi: 10.1007/s12094-021-02619-4. Epub ahead of print. PMID: 33840047. - Videtic GM, Paulus R, Singh AK, Chang JY, Parker W, Olivier KR, Timmerman RD, Komaki RR, Urbanic JJ, Stephans KL, Yom SS, Robinson CG, Belani CP, Iyengar P, Ajlouni MI, Gopaul DD, Gomez Suescun JB, McGarry RC, Choy H, Bradley JD. Long-term Follow-up on NRG Oncology RTOG 0915 (NCCTG N0927): A Randomized Phase 2 Study Comparing 2 Stereotactic Body Radiation Therapy Schedules for Medically Inoperable Patients With Stage I Peripheral Non-Small Cell Lung Cancer. Int J Radiat Oncol Biol Phys. 2019 Apr 1;103(5):1077-1084. doi: 10.1016/j.ijrobp.2018.11.051.)

Page 7, section 3.5.7. Combination with chemotherapy. The Authors could also cite the following paper (either in this section or in the Discussion section), which demonstrate better result of SBRT after Bevacizumab in lung oligometastases from colon cancer: Mazzola R, Tebano U, Aiello D, Paola GD, Giaj-Levra N, Ricchetti F, Fersino S, Fiorentino A, Ruggieri R, Alongi F. Increased efficacy of stereotactic ablative radiation therapy after bevacizumab in lung oligometastases from colon cancer. Tumori. 2018 Dec;104(6):423-428. doi: 10.5301/tj.5000701.

Page 7, line 199, 3.6. Prognostic factors of specific histologists: please, correct in "histologies".

Page 8, line 248-249, Discussion section. Please correct "To our knowledge, this PRISMA literature review is the largest study reported lung 248 SBRT for oligometastatic lesion" with "To our knowledge, this PRISMA literature review is the largest study regarding lung SBRT for oligometastatic lesion."

Page 8, line 271, Discussion section. Please replace "flying zone" with "no-fly zone".

Page 8, line 273, Discussion section. While talking about central lesions, please refer to Simultaneous Integrated Protection (SIP) strategies, citing also the following paper: "Mazzola R, Ruggieri R, Figlia V, Rigo M, Giaj Levra N, Ricchetti F, Nicosia L, Corradini S, Alongi F. Stereotactic body radiotherapy of central lung malignancies using a simultaneous integrated protection approach : A prospective observational study. Strahlenther Onkol. 2019 Aug;195(8):719-724. doi: 10.1007/s00066-018-01419-0."

Page 9, line 307, Discussion session: "carcinologic course" does not sound very well, probably "history of disease" would be better.

Page 9, line 309-312, Discussion session: "SBRT could be to extend the time until systemic treatment is needed for polymetastatic disease and thus maintain the patient's quality of life. In contrast, induced oligometastatic disease with patients with a history of polymetastatic disease having a disease with metastatic potential." The sentences are not clear.

Reviewer 3 Report

The study is well written and focuses on an interesting clinical issue the one of SBRT for lung metastatic lesions.  It has focused on appropriate studies published in the literature. The study well summarizes the clinical and prognostic concluding points from the published data.

Specific comments:

  1. Table 1 can go to supplemental data
  2. Table 2: try to provide max dimensions in cm uniformly.
  3. ‘Primitive cancer’: do you mean ‘Primary cancer’?
  4. ‘Metabolic imagery’: do you mean ‘Metabolic imaging’
  5. Try to provide a new table summarizing the clinical/prognostic factors
  6. Figure 2: BED in the figure and throughout the paper: Please mention the α/β value you used for calculations. Is it 10Gy?
  7. English syntax and grammar needs checking

Round 2

Reviewer 1 Report

Once again this is a review on a series of papers dealing with SBRT and lung metastases for oligometastatic disease from different primaries. They are presenting the data available in the different series and the key message SBRT is effective providing an adequate dose but we need more prospective studies with clear definition. This is a message already known and the paper add few new information but it is a comprehensive review of the papers available on the topic usin g their criteria for this review. 

Reviewer 3 Report

TABLE 2: It is better to omit those studies that provided no information on the clinical prognostic factors. Keep only those that report on this.